# Level of Knowledge Regarding Mpox among Peruvian Physicians during the 2022 Outbreak: A Cross-Sectional Study

**DOI:** 10.3390/vaccines11010167

**Published:** 2023-01-12

**Authors:** Jose A. Gonzales-Zamora, David R. Soriano-Moreno, Anderson N. Soriano-Moreno, Linda Ponce-Rosas, Lucero Sangster-Carrasco, Abraham De-Los-Rios-Pinto, Raysa M. Benito-Vargas, Valentina Murrieta-Ruiz, Noelia Morocho-Alburqueque, Brenda Caira-Chuquineyra, Daniel Fernandez-Guzman, Fabricio Ccami-Bernal, Carlos Quispe-Vicuña, Mariano Alarcon-Parra, Antony Pinedo-Soria, Jorge Alave

**Affiliations:** 1Division of Infectious Diseases, Department of Medicine, Miller School of Medicine, University of Miami, Miami, FL 33136, USA; 2Peruvian American Medical Society, Albuquerque, NM 87111, USA; 3Unidad de Investigación Clínica y Epidemiológica, Escuela de Medicina, Universidad Peruana Unión, Lima 15464, Peru; 4Facultad de Medicina, Universidad Peruana de Ciencias Aplicadas, Lima 15067, Peru; 5Escuela Profesional de Medicina Humana, Universidad Nacional de San Antonio Abad del Cusco, Cusco 08000, Peru; 6Sociedad Científica de Estudiantes de Medicina de la Amazonía Peruana (SOCIEMAP), Universidad Nacional de la Amazonía Peruana, Loreto 16000, Peru; 7Sociedad Científica de Estudiantes de Medicina de la Universidad Nacional de Piura (SOCIEMUNP), Piura 20002, Peru; 8Grupo Peruano de Investigación Epidemiológica, Unidad para la Generación y Síntesis de Evidencias en Salud, Universidad San Ignacio de Loyola, Lima 15012, Peru; 9Facultad de Medicina, Universidad Nacional de San Agustín de Arequipa, Arequipa 04000, Peru; 10Sociedad Científica de San Fernando, Universidad Nacional Mayor de San Marcos, Lima 15081, Peru; 11Hospital II ESSALUD Pucallpa, Ucayali 25001, Peru; 12Facultad de Medicina Humana, Universidad Nacional de San Martin, Tarapoto 22160, Peru; 13Departamento de Medicina, Facultad de Medicina, Universidad Peruana Unión, Lima 15464, Peru; 14Clinica Good Hope, Lima 15074, Peru

**Keywords:** monkeypox, knowledge, physicians, Peru (MeSH terms)

## Abstract

Introduction: Due to the high incidence of mpox in Peru and the poor knowledge about this disease among healthcare workers in non-endemic countries, it is crucial to determine the knowledge status of Peruvian physicians. Methodology: We conducted an analytical cross-sectional study based on an online survey from August to September 2022. Physicians who had a medical license and lived and practiced medicine in Peru were included. To evaluate the factors associated with a higher level of knowledge, we used crude (cPR) and adjusted (aPR) prevalence ratios with 95% confidence intervals (95% CI) using Poisson regression. Results: We included 463 physicians. The mean age was 36.6 (SD: 10.3) years, and most were male (58.1%). Regarding knowledge, the median knowledge score was 14 [IQR: 13 to 15] out of 17 points. In terms of knowledge gaps, only 60.7% of the participants knew that there was an FDA-approved vaccine for mpox, 49.0% of participants knew about mpox proctitis and 33.3% acknowledged that it could be transmitted by the bite of an infected rodent. We found that taking care of patients with mpox (aPR: 1.39; 95% CI: 1.13 to 1.72) was associated with higher knowledge (>p50), while living in the eastern macro-region (aPR: 0.62; 95% CI: 0.42 to 0.93) was associated with lower knowledge (≤p50). Conclusions: Our study showed a high level of knowledge about mpox among Peruvian physicians. However, educational campaigns may be necessary, especially for physicians from the eastern region and those who do not have clinical experience with mpox.

## 1. Introduction

Although the COVID-19 pandemic has not ended, mpox has emerged as an infectious disease that has gained a lot of relevance in recent months. Cases of mpox have been reported since the 1970s in several endemic countries in Africa; however, in May 2022 numerous cases appeared in non-African countries and have spread rapidly around the world [1]. Endemic mpox is usually self-limiting, with an average case-fatality rate of 8.7% [2]. Clinical manifestations include fever, prominent lymphadenopathy, and multiple papular or vesiculopustular lesions on the face and body. Complications develop mainly in immunocompromised individuals and children and include encephalitis, pneumonitis, and secondary bacterial infections [1].

Multiple infections need to be considered in the differential diagnosis of mpox, especially varicella, smallpox, herpes simplex virus, and other sexually transmitted infections. Varicella infection presents as a vesicular rash and compared to monkeypox, lacks lymphadenopathy and its lesions are typically in diverse stages of development [3]. Smallpox on the other hand, presents with very distinctive pox-like rash lesions but its occurrence would be extremely unlikely given the eradication of this disease in 1980. However, it would be important to consider in the context of bioterrorism [4].

The present outbreak in non-endemic countries has been much larger compared to previous ones, especially in regions of North America and Europe [2] and by 30 June 2022, more than 67,000 cases and 13 deaths had been reported in countries that historically had no cases [5]. This high incidence could have been related to factors such as biological changes in the virus, decreased immunity against smallpox, the resumption of international travels, among others [6].

In Latin America, the panorama has been very similar; by 28 June 2022, at least 48 confirmed cases and 16 suspected cases had been detected [7]; this posed a public health problem due to the lack of resources and measures to deal with mpox in this region [8]. Regarding Peru, the Ministry of Health (MINSA) confirmed the first case on 26 June 2022, in a foreigner [9].

The recent 2022 mpox outbreak is being transmitted person-to-person, whereas, in previous outbreaks, mpox was considered mainly a zoonosis [10]. The mode of transmission is thought to occur predominantly from direct contact with active skin lesions, especially during intimate or sexual contact, affecting mostly men who have sex with men (MSM). In fact, according to the Centers for Disease Control and Prevention (CDC), 99% of mpox cases have occurred in men, 94% of whom reported recent sexual or intimate contact with other men [11].

To date, mpox is not considered a sexually transmitted disease; however, it remains unknown whether asymptomatic patients can transmit the disease through body fluids during sexual intercourse. Other indirect routes of transmission have also been described, such as respiratory transmission through droplets or contaminated materials such as bed linens and towels [2,3,6,12]. There are still cases in which the source of infection has been difficult to trace, such as the case of a pediatric patient in the Netherlands who became infected without evident exposure to sick contacts [13], raising concerns about unrecognized modes of transmission and the possibility that mpox may evolve increasing its pathogenicity and transmissibility [14].

Adequate knowledge about an infectious agent, and its route of transmission plays a key role in infection-control planning and execution. However, the emergence of new infections constitutes a challenge for healthcare professionals because many of them may not have enough knowledge to diagnose and treat patients that present with a clinical picture they are not familiar with (knowledge gaps). All of which leads to inappropriate attitudes from healthcare personnel that can generate a negative impact on physicians’ decisions, prevention, and early detection of infections [15,16]. Therefore, it is crucial that surveillance systems take into account the level of knowledge of physicians in order to provide appropriate public health responses to limit the spread of emerging infections [17], such as the ongoing mpox outbreak [10].

Given the high incidence of mpox in Peru in the present outbreak and the poor knowledge about this infection among healthcare workers reported in other countries, it is important to determine the knowledge status of Peruvian physicians to identify, control, and manage cases appropriately [10]. The objective of this study was to evaluate the level of knowledge about mpox among Peruvian physicians and to determine the factors associated with higher knowledge. This study will make it possible to identify gaps in knowledge and to carry out educational campaigns aimed at reinforcing the concepts needed for adequate detection of cases.

## 2. Methodology

### 2.1. Study Design, Setting, and Participants

We conducted an analytical cross-sectional study based on an online survey from 10 August to 4 September 2022. We followed the STROBE guidelines for cross-sectional studies (Appendix A) [18]. In Peru, the first case of mpox was diagnosed on 26 June 2022. At the time of initiating data collection, the number of mpox cases was 632, and by the end of data collection exceeded 1500 cases [19]. For the present study, we included physicians who obtained a medical license in Peru, that were of Peruvian nationality, and were living in Peru at the time of the survey. Additionally, we considered for enrollment only physicians that were actively practicing medicine in the clinical, administrative, or research setting. We excluded those physicians who reported inadequate or incomplete information in the questionnaire.

### 2.2. Sample and Diffusion

Due to the exploratory nature of the study, we calculated the sample size based on a population of 98,290 licensed physicians in Peru [20]. We considered a conservative assumption that 50% of Peruvian physicians would have good knowledge, with a confidence level of 95% and a margin of error of 5%, obtaining a necessary sample of 385 participants. In addition, we assumed that 20% of the respondents did not meet the inclusion criteria or did not adequately complete the questionnaire, so we planned to send the survey to a minimum of 462 physicians. It was not possible to ensure that all physicians have access to electronic communication. For this purpose, we used a nonprobability convenience sampling. The questionnaire was distributed through the social networks Facebook, Twitter, Instagram, and WhatsApp.

### 2.3. Questionnaire

The survey was developed in Google Forms (Appendix A). We divided the questions into the following sections: (1) informed consent and inclusion criteria, (2) socio-demographic information, and (3) 17 knowledge questions (diagnosis, clinical manifestations, routes of transmission, treatment, and preventive measures) about mpox. For the knowledge questions, we took as a model a similar survey applied in Indonesia that aimed to assess the level of knowledge among general practitioners [10]. The knowledge questions were in Indonesia’s official language, Bahasa Indonesia. They were translated into English for publication in a journal. Subsequently, we used the English version to translate into Spanish, Peru’s official language. In addition, we adapted the questions to the Peruvian reality, incorporating concepts and data specific to the current epidemic outbreak. Likewise, this survey was validated through a review by experts in the field of infectious diseases in Peru and the United States of America.

### 2.4. Independent Variables

We considered the following variables as independent: gender (male, female); age; macro-region of residence; undergraduate university (public, private); years since graduation from medical school; medical position (general practitioner, resident or specialist); postgraduate degree (doctorate, master’s or none); area of work (urban, rural); awareness about the current mpox outbreak; and experience in taking care of a patient with mpox.

For the analysis, we considered 5 macro-regions in Peru (Lima and Callao, Center, North, East, and South) based on the geographic classification stipulated by the National Epidemiology Prevention and Infection Control Center of Peru (CDC-Peru) [21].

### 2.5. Dependent Variable: Knowledge of Mpox

Based on the knowledge questions, the minimum score was 0 and the maximum was 17 points; the higher the score, the greater the knowledge. To evaluate the factors associated with higher knowledge we dichotomized the variable according to whether the score was above the 50th percentile (higher knowledge) or below or equal to the 50th percentile (lower knowledge).

### 2.6. Statistical Analysis

We performed the statistical analysis in the R program version 4 (R Foundation for Statistical Computing, Vienna, Austria). We described categorical variables with relative and absolute frequencies, and numerical variables with mean and standard deviation (SD) or median and interquartile range (IQR) according to their normal or non-normal distribution, respectively. For bivariate analysis, we used Fisher’s test for nominal variables and the Wilcoxon rank sum test for numerical variables. To evaluate the factors associated with a higher level of knowledge about mpox we used crude (cPR) and adjusted (aPR) prevalence ratios with 95% confidence intervals (95% CI) using Poisson regression with robust variance. Variables with a *p*-value < 0.1 in the bivariate model were included in the adjusted models. A *p*-value < 0.05 was considered statistically significant.

### 2.7. Ethical Aspects

The present study was approved by the institutional ethics committee of the Universidad Peruana Unión (Approval 022-CEUPeU-011) and was registered in the Proyectos de Investigacion en Salud (PRISA, by its Spanish acronym) database of the Peruvian National Institute of Health. At the beginning of the survey, informed consent was requested from each participant, the survey was anonymous, and the data obtained were confidential.

## 3. Results

Out of a total of 477 respondents, eight did not meet the inclusion criteria and six were excluded due to incomplete information, resulting in a final sample of 463 participants (Figure 1). The mean age of the physicians was 36.6 (SD: 10.3) years, the majority were male (58.1%), and lived in Lima and Callao (56.8%). In addition, 56.6% did their undergraduate studies in a public university, with a mean time of graduation from medical school of 9.9 (SD: 9.5) years. Half of the respondents (50.5%) were general practitioners and 73.9% of them did not have a master’s degree or doctorate. Regarding the workplace, 43.5% reported working in a public hospital, 30.4% in private clinics, and 26.3% worked in a health center or health post. Likewise, 84.2% reported working in an urbanized area, and 14.5% reported having taken care of patients diagnosed with mpox. While most of the participants knew about the current outbreak, five physicians were not aware of the re-emergence of this disease (Table 1).

Regarding the participants’ sources of information on mpox, we found that 72.2% of physicians obtained information from the official websites of health institutions (MINSA, CDC, and World Health Organization [WHO]). Radio or television was reported as a source of information by 50.0% of participants. Social networks were used by 44.4% of physicians, whereas books and scientific journals were reported by 38.8%. Other sources of information such as conversations with colleagues, training courses, newspapers or magazines for the general public, conferences, and WhatsApp chain messages were less used (Figure 2).

Regarding knowledge of mpox, the median knowledge score was 14 [IQR: 13 to 15] out of 17 points. Almost all participants knew that the etiologic agent of mpox was a virus (99.6%), that mpox was reported in Peru (99.1%), and that it could be transmitted from person to person (98.7%). On the other hand, only 60.7% of the participants knew that there was a vaccine for mpox approved by the United States Food and Drug Administration (FDA), 49.0% of participants knew about mpox proctitis and 33.3% acknowledged that it could be transmitted by the bite of an infected rodent (Table 2).

The frequency of presenting a level of knowledge about mpox above the 50th percentile (>14/17 points) was 46.7%; with a higher proportion among those living in the Lima and Callao region (50.6%) and those living in the North region (54.5%), and those who had previously taken care of patients with mpox (62.7%) (Table 3). In addition, the majority (92.9%) of the physicians had more than 70% of the answers correct.

In the adjusted multivariate Poisson regression model, we found that taking care of patients with mpox (aPR: 1.39; 95% CI: 1.13 to 1.72) was independently associated with higher knowledge (>p50), while residing in the East macro-region (aPR: 0.62; 95% CI: 0.42 to 0.93) was associated with lower knowledge (≤p50) (Table 4).

## 4. Discussion

This study serves as the first report to demonstrate a high level of knowledge regarding mpox among Peruvian physicians during the present outbreak. Notably, the majority of participants (92.9%) had at least 70% of correct answers, with a median score of 14/17, which is an interesting finding given that mpox has never been reported in Peru before the current outbreak [22]. There have been other studies that evaluated mpox knowledge among healthcare professionals, such as the one conducted by Alsafani et al. in Kuwait, in which the study population was composed of nurses, pharmacists, medical technicians, and physicians. Among them, physicians obtained the highest knowledge score, with a mean of 4.6, indicative of good knowledge [23]. A study with a similar design was conducted in Jordan, where physicians also obtained high knowledge scores, only surpassed by medical technicians [24]. Other authors have reported lower levels of knowledge, such as Harapan et al., whose study revealed that 9.0% of physicians in Indonesia had good mpox knowledge when they used an 80% cutoff point, reaching a percentage of 36.5% when the cutoff was reduced to 70% [10]. On the other hand, Alshahrani et al. reported that 55% of physicians had good knowledge of mpox in Saudi Arabia, a level that was considered poor by the authors [25].

There are several factors that could have contributed to the high level of knowledge found in the present study. The time period elapsed from the first reported mpox case in Peru to the initiation of our study was approximately 1.5 months, which could have been enough time for the physicians to become familiar with the basic concepts of this disease [1,26]. It is important to note that the recruitment phase for our study (from 10 August to 4 September 2022) started when there were more than 500 mpox cases in Peru, exceeding 1500 cases by the end of data collection [27,28]. In addition, Peru was ranked seventh among the 10 countries most affected by mpox globally and ranked second behind Brazil in Latin America according to the WHO [29]. In this context, in July 2022, the Ministry of Health in Peru initiated an awareness-raising campaign in the community and among healthcare workers to limit the spread of mpox, a disease that at that point represented a public health emergency worldwide [24,30]. All these factors suggest that Peruvian physicians had constant exposure to information about the current mpox outbreak. In fact, our survey demonstrated that 99.1% of physicians were aware of the cases reported in Peru and only 0.4% of our sample stated that they did not receive any information about this infection. Another contributing factor was the frequent use of reliable sources of information among physicians, with 72.2% of participants obtaining information from official websites (WHO, CDC, etc.). Of note, the physicians who participated in our study were relatively young (mean age 36.6 years), and possibly more up to date on recent events given their constant access to online information through websites and social media [10,25]. Geographically, 56.9% of our population was from Lima and Callao (located on the coast), where most of the cases of mpox have been reported, and the majority (84.2%) of respondents were from urban areas, where information is more accessible. Finally, we believe that the relatively high percentage of physicians in residency training or who had a specialty (49.0%) in our study could have contributed to the high level of knowledge.

Regarding knowledge gaps in our population, our study revealed that 60.7% of the physicians knew about the mpox vaccine (Jynneos vaccine), an attenuated vaccine previously used for smallpox, which was adapted and approved by the FDA for mpox [31,32]. This rate was similar to that reported by Ricco et al. in Italy, where 60.1% of physicians acknowledged that an effective vaccine was available for mpox [33]. Other authors have reported a higher rate, with 69.8% of physicians that knew about a specific mpox vaccine according to a study from Saudi Arabia [25]. On the other hand, lower percentages have been reported in Jordan (33.3%), and Indonesia (36%) [24,34]. It is worth mentioning that by the time our study was conducted, only a few countries had started their vaccination campaign, and none of the Latin American countries had mpox vaccines available [35,36], all of which could have explained why almost 40% of our study population was not familiar with the vaccine.

In terms of knowledge about mpox symptoms, we found that only 49% of physicians recognized proctitis as a clinical manifestation of this disease. Traditionally, mpox lesions have been limited to the skin; however, during the present outbreak, affected patients have increasingly reported proctitis, a manifestation not commonly observed with mpox in previous years [37]. According to several reports, proctitis could be present in approximately 14% of the patients; however, its frequency has been reported to be as high as 25% in Spain [38], and it could be even the initial symptom in affected people [39,40,41]. We believe that it is crucial for clinicians to be aware of mpox proctitis to assure the proper identification, treatment, and isolation of cases, with the ultimate goal of limiting the spread of this disease.

Our study also found that 66.7% of physicians did not know that mpox could be transmitted by rodents, which is striking given that mpox has been historically described as a zoonosis in endemic countries, where the primary hosts are rats, squirrels, and other rodents [42]. However, the community spread of mpox in non-African countries during the present outbreak has occurred exclusively by human-to-human transmission, and mainly via direct contact with skin lesions during sexual intercourse [37]. It is important to mention that the most affected population has been MSM, with the highest risk among those with multiple sexual partners, participating in mass gatherings, and with a history of sexually transmitted diseases [37,43,44]. Transmission from rodent to human has not occurred outside Africa during the present outbreak, which can potentially explain the poor knowledge about this mode of transmission among Peruvian physicians.

Our study also assessed the factors associated with higher knowledge; in this regard, taking care of a patient with mpox was associated with higher knowledge. A reasonable explanation is that exposure to real scenarios and active learning are fundamental to develop good knowledge, critical thinking, better perception, and cognition towards a disease in the medical setting [45,46]. We believe that educational campaigns should contemplate experiential learning for healthcare workers to develop better and sustainable outcomes [47,48]. Considering that the physicians with clinical experience have a higher level of knowledge, it would be highly recommended that these professionals take the lead in educating their colleagues and community about mpox, to improve the detection rate and management of this disease.

Interestingly, our study also revealed that living in the East macro-region, which is predominantly jungle, was associated with lower knowledge. A possible reason is the limited access to information in this part of the country, where at times there is no good internet connection, especially in rural and low-income areas. Another factor that could explain this finding is that most mpox cases have occurred in the Lima and Callao macro-region, where 79% of the cases have been found. Only 1.4% of patients have been reported from the East macro-region [49], which could explain why physicians in this region were less familiar with this infection. However, it is also possible that the low prevalence of mpox in the East macro-region might have been due to the poor recognition and inadequate diagnosis of the affected patients by healthcare workers, for which implementation of educational campaigns may be needed.

Other authors have identified additional factors associated with higher knowledge. In this regard, Harapan et al. found that good knowledge was associated with age under 30 years, graduation from universities located in Java, Indonesia, and working in community health centers [10]. Furthermore, Alshahrani et al. demonstrated that good knowledge was also associated with the following factors: female gender; being a general practitioner; working in the private sector; and receiving information about mpox during medical schools or residency years [25]. Our study did not reveal an association with age, location, or workplace. Another study aimed at evaluating knowledge in Kuwait found that lower knowledge was associated with the false notion that mpox was exclusive to male homosexuals, and with a higher embrace of conspiracy beliefs regarding emerging viral infections [23]. However, this study evaluated a different population that included not only physicians, but also nurses, pharmacists, and medical technicians. A similar study in Jordan also revealed that lower knowledge was associated with conspiracy theories about virus emergence [21]; unfortunately, this factor was not evaluated in the present study.

Notably, a study conducted in the Czech Republic revealed interesting findings about the level of mpox-related knowledge and vaccine hesitancy among Czech healthcare workers. They analyzed demographics (sexual orientation, marital status, and having minors) as well as medical anamnesis (e.g., chronic medical conditions and COVID-19 vaccination status) related to mpox vaccine rejection, hesitancy and acceptance. These particular variables were not evaluated in our study regarding level of mpox knowledge and vaccine acceptability. As in our study, Czech healthcare workers demonstrated suboptimal levels of knowledge regarding the availability of effective vaccines (33.7%) and antivirals (25.2%) for mpox. Vertical transmission was one of the least correctly answered items by Czech healthcare workers (23.5%). This item was not evaluated in the present study. In contrast to our study, scientific journals (5.6%) and the U.S. CDC (1.5%) were the least frequent sources of information among Czech healthcare workers, which highlights the systemic problem of the suboptimal practice of evidence-based medicine in the Czech Republic and its impact on epidemic awareness and misinformation, all of which has a negative impact in mpox vaccine acceptance among Czech healthcare workers, reaching an acceptance rate of only 8.8% [50].

There are some limitations in our study. Our sample was not completely representative of the entire physician population in Peru. Given that the study was based on an online survey, only physicians with internet access had the possibility to participate, resulting in a selection bias. Furthermore, the mean age was 36.6 years corresponding to young physicians who are usually more familiar with web-based surveys, resulting in their higher participation in the study [51]. Another limitation was the possibility of dishonesty in answering the questions (looking up information, sharing answers, etc.), a factor that could not be completely excluded due to the lack of participants’ supervision during the survey completion.

Despite these limitations and its exploratory nature, to our knowledge, the present study is the first one to assess the level of knowledge about mpox in physicians in Latin America during the 2022 outbreak and hopefully can serve as a model for future studies. In addition, our study involved a sample of more than 400 Peruvian physicians from different regions, which makes it the largest study of its kind published to date [23,24,25,33,52].

## 5. Conclusions

Our study showed a high level of knowledge about mpox among Peruvian physicians. However, there are still some important knowledge gaps among physicians, such as the relatively poor knowledge about mpox proctitis, FDA-approved vaccine, and zoonotic nature of this infection. We also found that clinical experience with mpox is associated with higher knowledge whereas living in the East macro-region (predominantly jungle) is associated with lower knowledge, suggesting that educational campaigns might be needed in eastern Peru to improve detection of cases. We also advocate for physicians with clinical experience to lead mpox educational activities in the community and among their peers.

## Figures and Tables

**Figure 1 vaccines-11-00167-f001:**
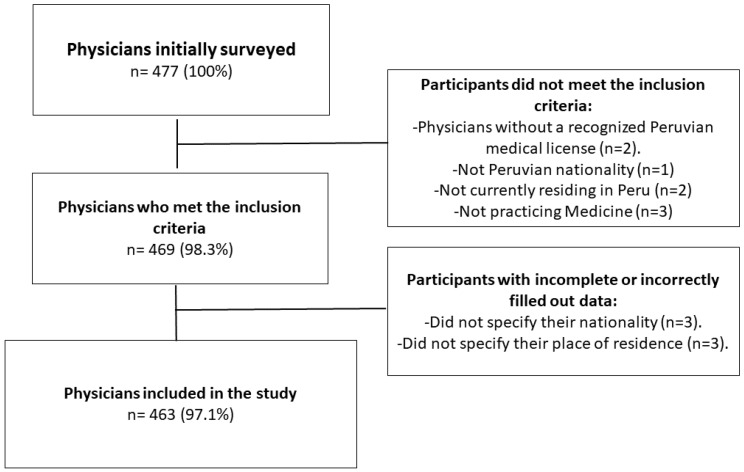
Flowchart of the participants’ selection. n: number.

**Figure 2 vaccines-11-00167-f002:**
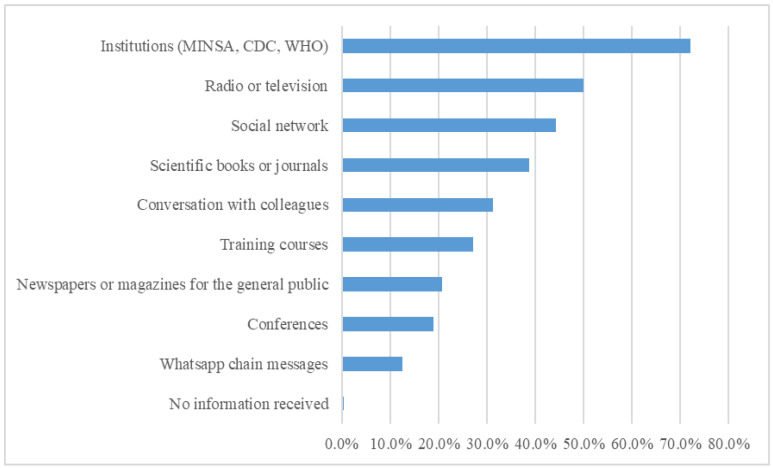
Sources of information on mpox used by Peruvian physicians.

**Table 1 vaccines-11-00167-t001:** Characteristics of the participants.

Characteristic	N = 463
Gender	
Female	194 (41.9%)
Male	269 (58.1%)
**Age, mean (SD)**	36.6 (10.3)
**Macro-region**	
Lima and Callao	263 (56.8%)
Central	27 (5.8%)
North	44 (9.5%)
Eastern	58 (12.5%)
South	71 (15.3%)
**Undergraduate University**	
Private University	201 (43.4%)
Public University	262 (56.6%)
**Years from medical school graduation, mean (SD)**	9.9 (9.5)
**Medical position**	
Medical Specialist	189 (40.8%)
General Practitioner	234 (50.5%)
Resident Physician (in training)	40 (8.6%)
**Postgraduate degree**	
Doctorate	4 (0.9%)
Master’s degree	117 (25.3%)
None	342 (73.9%)
**Area of work**	
Rural Zone	73 (15.8%)
Urban Zone	390 (84.2%)
**Awareness about the current mpox virus**	
No	5 (1.1%)
Yes	458 (98.9%)
**Experience in taking care of a patient with mpox**	
No	396 (85.5%)
Yes	67 (14.5%)
**Knowledge score, median (IQR)**	14.00 (13.0, 15.0)
**Knowledge percentile**	
≤p50, 0–14 points	247 (53.3%)
>p50, >14 points	216 (46.7%)
**% of correct answers**	
<70%	33 (7.1%)
>70%	430 (92.9%)

SD—standard deviation; IQR—interquartile range; p50—50th percentile.

**Table 2 vaccines-11-00167-t002:** Proportion of correct answers by question.

Questions: Correct Answer	N = 463	
Mpox is caused by: A virus	99.6%	461
Cases of mpox have been reported in Peru: True	99.1%	459
Mpox can be transmitted from person to person: True	98.7%	457
What treatment would you use for people with mild symptoms of mpox? Symptomatic treatment.	96.5%	447
Mpox has a mortality rate of approximately 80%: False	96.1%	445
People with symptoms of mpox require isolation to limit the spread of the infection: True	95.0%	440
Vesicles are one of the classic lesions of mpox: True	91.6%	424
Patients with mpox may present with influenza-like symptoms in the first few days of illness: True	90.9%	421
It is not a route of transmission of mpox: Mosquito bite	89.6%	415
What is the incubation period of mpox? 5–21 days	89.4%	414
Mpox is a disease endemic to Central and West Africa: True	86.0%	398
Mpox lesions do not occur on the genitals: False	83.8%	388
What is the population most affected by mpox in the present outbreak? Men who have sex with men	83.6%	387
Which test is used to confirm the diagnosis of mpox? PCR of the lesions	81.4%	377
There is an FDA-approved vaccine to prevent mpox: True	60.7%	281
Proctitis is one of the manifestations of mpox: True	49.0%	227
Mpox can be transmitted to a person by the bite of an infected rodent: True	33.3%	154

**Table 3 vaccines-11-00167-t003:** Comparison of characteristics between those physicians with higher knowledge (>p50) and those with lower knowledge (≤p50) of mpox.

Characteristic	≤P50,	>P50,	cPR [95% CI] *	*p*-Value
	N = 247	N = 216		
**Gender**				
Female	109 (56.2%)	85 (43.8%)	Ref.	Ref.
Male	138 (51.3%)	131 (48.7%)	1.11 [0.91 to 1.36]	0.301
**Age, mean (SD)**	36.4 (10.2)	36.8 (10.5)	1.00 [0.99 to 1.01]	0.678
**Macro-region**				
Lima and Callao	130 (49.4%)	133 (50.6%)	Ref.	Ref.
Central	18 (66.7%)	9 (33.3%)	0.66 [0.38 to 1.14]	0.093
North	20 (45.5%)	24 (54.5%)	1.08 [0.80 to 1.45]	0.060
East	40 (69.0%)	18 (31.0%)	0.61 [0.41 to 0.92]	0.007
South	39 (54.9%)	32 (45.1%)	0.89 [0.67 to 1.18]	0.415
**Undergraduate University**				
Private University	116 (57.7%)	85 (42.3%)	Ref.	Ref.
Public University	131 (50.0%)	131 (50.0%)	1.18 [0.97 to 1.45]	0.101
**Years from medical school graduation, mean (SD)**	9.64 (9.13)	10.2 (9.86)	1.00 [0.99 to 1.01]	0.488
**Medical position**				
Medical Specialist	97 (51.3%)	92 (48.7%)	Ref.	Ref.
General Practitioner	130 (55.6%)	104 (44.4%)	0.91 [0.74 to 1.12]	0.388
Resident Physician (in training)	20 (50.0%)	20 (50.0%)	1.03 [0.73 to 1.45]	0.880
**Postgraduate degree**				
Doctorate	3 (75.0%)	1 (25.0%)	Ref.	Ref.
Master’s degree	67 (57.3%)	50 (42.7%)	1.71 [0.31 to 9.45]	0.544
None	177 (51.8%)	165 (48.2%)	1.93 [0.35 to 10.6]	0.415
**Area of work**				
Rural Zone	42 (57.5%)	31 (42.5%)	Ref.	Ref.
Urban Zone	205 (52.6%)	185 (47.4%)	1.12 [0.84 to 1.49]	0.439
**Awareness about the current mpox outbreak**				
No	3 (60.0%)	2 (40.0%)	Ref.	Ref.
Yes	244 (53.3%)	214 (46.7%)	1.17 [0.40 to 3.43]	0.794
**Experience in taking care of a patient with mpox**				
No	222 (56.1%)	174 (43.9%)	Ref.	Ref.
Yes	25 (37.3%)	42 (62.7%)	1.43 [1.15 to A1.77]	0.005

* GLM Poisson with robust variance. SD—standard deviation; IQR—interquartile range; p50—50th percentile; Ref.—reference, cPR—crude prevalence ratio. The numbers in bold indicate that the *p*-value has achieved statistical significance (<0.05).

**Table 4 vaccines-11-00167-t004:** Multivariate analyses of the factors associated with a higher level of knowledge about mpox.

Variables	aPR *	CI 95%		*p*
**Macro-region**				
Lima and Callao	Ref.			
Central	0.67	0.39	1.15	0.148
North	1.09	0.81	1.46	0.566
East	0.62	0.42	0.93	0.022
South	0.91	0.69	1.20	0.504
**Experience in taking care of a patient with mpox**				
No				
Yes	1.40	1.13	1.73	0.002

* GLM Poisson with robust variance. aPR—adjusted prevalence ratio, CI—confidence interval. The numbers in bold indicate that the *p*-value has achieved statistical significance (<0.05).

## Data Availability

Research data is not available due to privacy.

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
