# Peer review of "Level of Knowledge Regarding Mpox among Peruvian Physicians during the 2022 Outbreak: A Cross-Sectional Study"

_vaccines, 2023, doi:10.3390/vaccines11010167_

Round 1

Reviewer 1 Report

Dear Author(s),

Thank you for your esteemed efforts in increasing our collective knowledge about the knowledge level of monkeypox (mpox) among medical physicians in Peru.

1. Please use the official name (mpox) in the entire manuscript.

2. The manuscript requires language proofreading and editing.

3. The keywords should be better suggested according to the MeSH/NCBI: https://www.ncbi.nlm.nih.gov/mesh/

4. In the Introduction section, you may consider reflecting on the differential diagnosis of mpox - especially when compared to smallpox and chickenpox.
Suggested ref (optional):
https://onlinelibrary.wiley.com/doi/10.1002/jmv.28091

5. The Introduction section should define what the "knowledge gap" is. 

6. The study should be reported according to the STROBE guidelines for cross-sectional studies. Please cite the STROBE guidelines in your Methods section.
Suggested ref (optional):
https://doi.org/10.1016/S0140-6736(07)61602-X

7. Please upload the STROBE checklist as a supplementary file.
https://www.strobe-statement.org/checklists/

8. How did you ensure that the participants were really working as physicians?

9. How the Spanish version was produced?

10. What are the psychometric properties of the used questionnaire?

11. Do you have information about the specialisations of the participating physicians? The type of speciality may play a role in knowledge level.

12. The Discussion section may benefit from reflecting on additional similar studies.
Suggested ref (optional):
https://pubmed.ncbi.nlm.nih.gov/36560432/

Sincerely,

Author Response

Dear reviewer,

All the authors analyzed the suggestions and comments about the manuscript. To make the review easier we are using the “track changes” function of MS Word/LaTeX suggested. Therefore, any changes can be easily viewed by the editors and reviewers.

The answers to the queries are the following:

Q1. Please use the official name (mpox) in the entire manuscript.

A1. We changed “monkeypox” for “mpox” in the entire manuscript.

Q2. The manuscript requires language proofreading and editing.

A2. We did proofreading already.

Q3. The keywords should be better suggested according to the MeSH/NCBI: https://www.ncbi.nlm.nih.gov/mesh/

A3. The keywords were chosen according to the MeSH/NCBI.

Q4. In the Introduction section, you may consider reflecting on the differential diagnosis of mpox - especially when compared to smallpox and chickenpox.
Suggested ref (optional):
https://onlinelibrary.wiley.com/doi/10.1002/jmv.28091

A4. Differential diagnosis of mpox, especially smallpox and chickenpox were added to the second paragraph of the introduction.

Q5. The Introduction section should define what the "knowledge gap" is. 

A5. We added the definition of “knowledge gap” in the seventh paragraph of the introduction

Q6. The study should be reported according to the STROBE guidelines for cross-sectional studies. Please cite the STROBE guidelines in your Methods section.
Suggested ref (optional):
https://doi.org/10.1016/S0140-6736(07)61602-X

A6. We added the suggested reference in the first paragraph of methods.

Q7. Please upload the STROBE checklist as a supplementary file.
https://www.strobe-statement.org/checklists/

A7. We appreciate your comments. We added STROBE checklist as a supplementary material

Q8. How did you ensure that the participants were really working as physicians?

A8. Non-probability convenience sampling was used and the survey was anonymous, so the physician's identification was not requested. However, the questionnaire was distributed through social networks such as Facebook, Twitter, Instagram and WhatsApp, in groups belonging to the study researchers' environment, related to physicians. Likewise, the survey was distributed in the different departments of medicine in public hospitals, private clinics and health centers.

Q9. How the Spanish version was produced?

A9. We added the translation process: “The knowledge questions in the Monkeypox study we relied on were in Indonesia's official language, Bahasa Indonesia. They were translated into English for publication in the journal. Subsequently, we used the English version to translate it into Spanish, Peru's official language. In addition, we adapted the questions to the Peruvian reality, incorporating concepts and data specific to the current epidemic outbreak. Likewise, this survey was validated through the review of experts in the field of Infectious Diseases in Peru and the United States of America”

Q10. What are the psychometric properties of the used questionnaire?

A10. The survey was validated by 6 experts in the area of infectious diseases (2 from Peru and 4 from the USA); however, a Cronbach's alpha evaluation could not be performed because it does not apply to scales with dichotomous responses.

https://pubmed.ncbi.nlm.nih.gov/18940100/

Q11. Do you have information about the specialisations of the participating physicians? The type of speciality may play a role in knowledge level.

A11. We included the question on medical specialty in the sociodemographic variables section of the questionnaire: "What is your specialty in Medicine?". However, we found that the number of medical specialists was low, which did not allow us to perform a good analysis of the data.

Q12. The Discussion section may benefit from reflecting on additional similar studies.
Suggested ref (optional):
https://pubmed.ncbi.nlm.nih.gov/36560432/

A12. We added in the discussion a reflection about the interesting study suggested from Czech Republic which evaluate mpox-related knowledge and vaccination perceptions among healthcare workers. It is important to mentioned that we found some similarities and differences as well to the other studies from Indonesia, Italy, Jordan, Kuwait and Saudi Arabia previously described.

Reviewer 2 Report

·         Abstract and Elsewhere

Two geographic location designation approaches

Is Jungle defined in any way? And elsewhere, similarly undefined designations of coast and highlands appear. While it is somewhat obvious what these mean, they should be defined. For example, where do coast and jungle meet? Or Jungle and Highlands? This could potentially be based on locations and landcover data as found at https://www.usgs.gov/media/images/south-america-land-cover-characteristics-data-base-version-20. Or, perhaps census data are more appropriate and the use of the rural vs urban designations instead of the ambiguous jungle designation could help? You seem to analyze the geographic question with both a three-level jungle, coast, highlands approach and a two-level rural vs urban one. Are these redundant? Regardless, these should be described and the utility of including both explained in the methods section. Additionally, if you feel both are useful then comparing them in your results would seem important as would discussing those differences or similarities.  Either way, given the importance of some of these designations to your results, defining them and justifying the use of both the three-level and two-level geographic categories would seem important. 

·         2.1. Study design, setting, and participants

Are these physician groups analyzed to be certain their within-group responses are similar before lumping them together? Also-researchers would seem likely to be more highly represented in urban settings and might represent an undetected source of bias to this analysis as well.

·         2.2. Sample and diffusion

Can you be sure that the groups of physicians under investigation have equal access to these online resources? If not, the exclusion of more traditional non-electronic means of communication may represent unintended selection bias.

·         Results text and Figure 1.

Although the reasons for excluding eight physicians from the study are represented in Figure 1, it seems six additional physicians were excluded for reasons which are not made clear in your Results section and in Figure 1.

Author Response

Dear reviewer,

All the authors analyzed the suggestions and comments about the manuscript. To make the review easier we are using the “track changes” function of MS Word/LaTeX suggested. Therefore, any changes can be easily viewed by the editors and reviewers.

The answers to the queries are the following:

 Abbreviations: Query 1: Q1   Answer 1: A1

Q1. Abstract and Elsewhere

Two geographic location designation approaches

Is Jungle defined in any way? And elsewhere, similarly undefined designations of coast and highlands appear. While it is somewhat obvious what these mean, they should be defined. For example, where do coast and jungle meet? Or Jungle and Highlands? This could potentially be based on locations and landcover data as found at https://www.usgs.gov/media/images/south-america-land-cover-characteristics-data-base-version-20. Or, perhaps census data are more appropriate and the use of the rural vs urban designations instead of the ambiguous jungle designation could help? You seem to analyze the geographic question with both a three-level jungle, coast, highlands approach and a two-level rural vs urban one. Are these redundant? Regardless, these should be described and the utility of including both explained in the methods section. Additionally, if you feel both are useful then comparing them in your results would seem important as would discussing those differences or similarities.  Either way, given the importance of some of these designations to your results, defining them and justifying the use of both the three-level and two-level geographic categories would seem important. 

 A1. Thank you for your comment. We have changed the categories of the variable region of residence. For this we consulted the National Epidemiology Center of Peru (CDC-Peru) and they provided us with the classification of macro regions used for outbreaks in Peru. This classification divides the regions into 5 macro-regions: Lima and Callao, center, north, east and south. Regarding the area of work variable, it was self-reported and with the category urban and rural. It is different from the place of residence variable since, for example, within the jungle region there are urban and rural areas. We added a methods section entitled "independent variables": "We considered the following variables as independent: gender (male, female), age, macro-region of residence, undergraduate university (public, private), years since graduation from medical school, medical position (general practitioner, resident or specialist), postgraduate degree (doctorate, master's or none), area of work (urban, rural), awareness about the current mpox outbreak and experience in taking care of a patient with mpox.

For the analysis, we considered 5 macro-regions in Peru (Lima and Callao, Center, North, East, and South) based on the geographic classification stipulated by the National Epidemiology, Prevention and Infection Control Center of Peru (CDC-Peru).”

       Q2. Study design, setting, and participants

Are these physician groups analyzed to be certain their within-group responses are similar before lumping them together? Also-researchers would seem likely to be more highly represented in urban settings and might represent an undetected source of bias to this analysis as well.

A2. We appreciate the observation. It was not possible to analyze the physicians' responses according to these groups because it was a single general question that was included at the beginning of the questionnaire as an inclusion criterion, but not as a study variable. The original question was this:

“Are you currently practicing medicine (including attending, research or administrative positions)?”

  1. No (survey ends)
  2. Yes

      Q3. Sample and diffusion

Can you be sure that the groups of physicians under investigation have equal access to these online resources? If not, the exclusion of more traditional non-electronic means of communication may represent unintended selection bias.

 A3. Since not all physicians have access to traditional electronic equipment, we opted for a non-probabilistic convenience sampling to reduce the aforementioned bias.

  • Q4. Results text and Figure 1.

Although the reasons for excluding eight physicians from the study are represented in Figure 1, it seems six additional physicians were excluded for reasons which are not made clear in your Results section and in Figure 1.

A4. We added the reasons for exclusion in the first paragraph of the methodology. "We excluded those physicians who reported inadequate or incomplete information in the questionnaire."

Additionally, we corrected Figure 1 to display all the exclusion ratios with their respective frequencies.

Round 2

Reviewer 1 Report

Dear Authors,

Thank you for your esteemed efforts in responding to my previous comments and amending the manuscript accordingly.

Sincerely,